# Patients' perceived quality of care and their satisfaction with care given for MDR-TB at referral hospitals in Ethiopia

**Mengistu K. Wakjira**[1]*, **Peter T. Sandy**[2], **A. H. Mavhandu-Mudzusi**[3]

**1** Abt Associates Inc Ethiopia, Addis Ababa, Ethiopia, **2** College of Nursing, Midwifery & Healthcare, University of West London (Professor Extraordinarius, Development Studies, UNISA), London, United Kingdom, **3** College of Human Sciences, Office of Graduate Studies, and Research, University of South Africa, Pretoria, South Africa

* mengistukenea@gmail.com

**Data Availability Statement:** Data that support the findings of this study are uploaded on the online submission link. The authors hereby confirm that there is no any restriction on use of Qualitative data

## Abstract

### Background

There is presently dearth of evidence in Ethiopia on patients' perception on quality of care given for multi-drug resistant tuberculosis (MDR-TB) and their satisfaction with the care and services they receive for the disease. Moreover, there is no evidence on the experiences and practices of caregivers for MDR-TB regarding the functionality of the programmatic management of MDR-TB at referral hospitals in Ethiopia. Thus, this study was conducted to address these gaps. Evidence in these areas would help to institute interventions that could enhance patient satisfaction and their adherence to the treatment given for MDR-TB.

### Design and methods

This study employed an inductive phenomenological approach to investigate patients' perception of the quality of care given for MDR-TB, level of their satisfaction with the care they received for MDR-TB and the experiences and practices of caregivers for MDR-TB on the functionality of the programmatic management of MDR-TB at referral hospitals in Ethiopia. The data were analysed manually, and that helped to get more control over the data.

### Results

The majority of the patients were satisfied with the compassionate communication and clinical care they received at hospitals. However, as no doctor was dedicated exclusively for the MDR-TB centre of the hospitals, patients could not get timely medical attention during emergent medical conditions. Patients were dissatisfied with the poor communication and uncaring practice of caregivers found at treatment follow-up centres (TFCs). Patients perceived that socio-economic difficulties are both the cause of MDR-TB and it has also challenged their ability to cope-up with the disease and its treatment. Patients were dissatisfied with the poor quality and inadequate quantity of the socio-economic support they got from the programme. Despite the high MDR-TB and HIV/AIDS co-infection, services for both diseases were not available under one roof.

uploadad with the paper. PLOS has the write the make the qual Data available in approperiate respository. The authors also clarify that clarify the Qual Data "Qualitative_data_18_patientsn_11_Caregivers. doc.docx" can be published as a Supporting information file!!

**Funding:** During the time the research was conducted one of the authors was employed to Abt Associates Inc. operating in Ethiopia. The funder provided support in the form of salaries for authors (MK Wakjira) but did not have any additional role in the study design, data collection and analysis, decision to publish, or preparation of the manuscript. The specific roles of these authors are articulated in the 'authors contributions section.

**Competing interests:** During the time the research was conducted one of the authors was employed to Abt Associates Inc. operating in Ethiopia. The authors hereby confirm that this does not alter our adherence to PLOS ONE policies on sharing data and materials.

## Conclusions

Socio-economic challenges, inadequate socio-economic support, absence of integrated care for MDR-TB and HIV/AIDS, and the uncaring practice of caregivers at treatment follow-up centres are found to negatively affect patients' perceived quality of care and their satisfaction with the care given for MDR-TB. Addressing these challenges is recommended to assist patients' coping ability with MDR-TB and its treatment.

## Introduction

Ethiopia is among the 30 countries in the world described by the World Health Organization (WHO) as high burden for TB and MDR-TB. Thus, the government rolls out community-based programmatic management of drug-resistant TB across all its provinces [1]. Yet, there is lack of evidence on patients' perception on the quality of care given for MDR-TB and their satisfaction with the care and services they receive for the disease. Patients' perception of the quality of healthcare and their satisfaction with the care they receive impact both the process and the outcomes of the treatment given for MDR-TB. Health care quality is the degree to which health services increase the likelihood of desired health outcomes [2]. Attributes of healthcare quality include the knowledge of caregivers to deliver the promised care, willingness to attentively help patients, interpersonal communication, patient centredness and timeliness of the care and the quality of the physical facilities in which care is given [3–6]. The status of healthcare quality determines patients' perceptions about the quality of healthcare and their satisfaction with the care they receive [7]. Perceived quality is the patients' judgment about the overall excellence of the healthcare they receive [8]. Patients' perception of the quality healthcare and status of their satisfaction with the care they receive are measures of healthcare quality [9]. It measures the discrepancy between patient's expectations and their perception of the services given by an institution or by healthcare givers [10, 11]. Patients' satisfaction is patient-reported measure of treatment outcome [12]. Patient satisfaction is determined by patients' expectations regarding the services that they want to see in a health institution and their perception of the actual services that they receive. For example, patients' perceptions that their physician encounters are patient-centered are associated with better recovery from discomfort [13, 14]. For patients with MDR-TB, there are extra clinical factors that determine patient satisfaction with the care they receive and their ability to cope with the disease. These include socio-economic difficulties including lack of education, unemployment, poor living conditions, and under nutrition associated with financial constraints [15, 16]. MDR-TB inequitably affects the poor in resource-constrained regions of the world. In some settings, patients lost up to 40% of their annual discretionary income to MDR-TB. Patients also suffer from MDR-TB associated stigmatisation leading to low self-esteem, insults, ridicule, and social exclusion including termination of employment [17]. These result in psychological stress, decreased quality of life, and social status [18–21]. On the other hand, the treatment given for MDR-TB is lengthy, less effective, and poorly tolerated mainly associated with adverse effects of the drugs used to treat it [22–24]. In this regard, there is presently very limited evidence in Ethiopia on factors determining patients' perceived quality of care and level of their satisfaction with the care they receive for MDR-TB and the experiences and practices of caregivers for MDR-TB regarding the management of patients with MDR-TB. Lack of evidence has motivated the investigators to play a catalytic role in generating evidence that can help institute evidence-informed decisions in the programmatic management of MDR-TB.

## Methods and materials

An inductive phenomenological approach was used to understand the meanings that patients with MDR-TB and their caregivers attribute to their experiences and to the external world surrounding the treatment given for MDR-TB [25].

### Study setting and period

The study was conducted at Adama and Nekemte Referral Hospitals in the Oromia region of Ethiopia. The two hospitals were selected based on their long-years of experience on programmatic management of MDR-TB. The study was conducted between the 10th of November 2016 and 7th of February 2017 at the two hospitals.

### Study population and participant selection

Laboratory confirmed patients with MDR-TB who were enrolled to treatment for MDR-TB were the study population for this study. Data were collected from presumably information rich patients aged 18 years and above and from caregivers of patients with MDR-TB. Participants were selected using purposeful sampling methods to elicit their experience of this condition. From total of 136 patients who were on treatment for at least six months at time of data collection and those aged 18 years and above (study inclusion criteria), 23 information rich patients were identified by the help of MDR-TB focal points of the two hospitals. Patients aged less than 18 years and those who attended treatment for less than 6 months were excluded from the study. From total eligible patients, successful in-depth interviews were conducted with 18 patients, while 3 patients did not volunteer to participate and the interviews with 2 patients was not successful. Likewise, 11 active caregivers for patients with MDR-TB participated in the interviews with caregivers. The number of participants of the interviews was determined by category saturation, the point at which the interviews did not reveal new data relevant to the objectives of the study.

### Ethical considerations

For this study ethical clearance was obtained from University of South Africa (UNISA) and the two target hospitals were accessed through ethical clearance obtained from Oromia Health Bureau, Health Research Core Process. Access to patients with MDR-TB was obtained based on permission from the Chief Executive Officers of the two hospitals and from the caregivers for patients with MDR-TB [26]. Data were collected from patients aged 18 years who could understand the objectives of the study and can give their informed consent. Anonymous tools were used to collect data and no data were analysed in connection with participants' identifiers, instead codes were used to quote excerpts from patients and their caregivers.

### Data collection

A semi-structured in-depth interview guide was used to collect data. The interview guide was pilot tested on a sample of patients with MDR-TB and their caregivers found at another hospital similar to the two hospitals selected for the study. Data were collected by the principal investigator through face-to-face interviews with patients and their caregivers. Data were collected on patients' socio-demographic and socio-economic status, and patients' perceptions on socio-economic impact of becoming a patient with MDR-TB. Moreover, data were collected on the perceptions of patients on the quality of care given for MDR-TB and their satisfaction with the care they receive for the disease; including patient-centredness of the care, communication between patients and their caregivers, and the status of socio-economic

support provided by the programme of MDR-TB. Likewise, data were collected on caregivers' experience and practices regarding the functionality of the MDR-TB programme in providing the continuum of care needed by patients with MDR-TB, and the availability of integrated care for patients affected by both MDR-TB and HIV/AIDS. Data were also collected on status of community awareness about MDR-TB and level of system's support to the programme in coordinating the non-clinical aspects of the programmatic management of MDR-TB. Interview data were audio recorded and transcribed verbatim. Notes were taken during interviews where data were captured on the feelings and experiences of the study participants. Data from multiple sources were used to make sure that emerging themes are established based on converging different sources of data or different perspectives of the study participants.

## Data analysis

Data were analysed thematically, and that helped to capture nuances or latent meanings and get more control over the data. A focus was made on making sure that participants' subjective meanings are appropriately conveyed. For this, member checking was employed with each participant in which summarized interview data were presented to each interviewee to understand if what the researcher captured matches the intentions and opinions of the participants [27]. Transcripts were also given to literate participants, and they were asked to read and comment if what the researcher captured matches their opinions. Moreover, two debriefers who had experience in social research were located and their frequent feedbacks were used to enhance accuracy of the construct under scrutiny [28]. Meanings are conveyed in terms of themes and their related constituents or subthemes [29]. A theme was used to unify ideas at the interpretive level, and it helped to answer the study questions. Sub-themes helped to obtain a comprehensive view of the data and uncover patterns in the participants' accounts. As such, data analysis generated 29 sub-themes that were clustered under 7 major themes as briefly mentioned in the result section.

## Results

A total of eighteen adult patients with MDR-TB; 9 (50%) female and 9 (50%) male were included in the study. Three of the 18 (17%) participants openly disclosed that they were patients with MDR-TB and HIV co-infection. Moreover, eleven caregivers for MDR-TB (3 physicians and 8 nurses) participated in the in-depth interviews.

## Patients' perceived quality of care and their satisfaction with the care given for MDR-TB

Patient to provider communication, patients' engagement in treatment decision making, and caregivers' responsiveness to patients' emergent medical conditions were among the sub-themes related to this main theme. At the hospitals, patients were counselled on the treatment given for MDR-TB and on the importance of treatment adherence. There was smooth communication between patients and their hospital level caregivers. Patients had a good perception of the smooth communication and medical services given by caregivers at hospitals. Hospital level caregivers were described by patients as empathetic and caring. Thus, patients were satisfied with the clinical care they received at the hospitals. The next excerpt illustrates this:

> "....the doctor has been suffering with me and he even pays for my transport from his own pocket and also gives me money for my lunch...

[female participant, number eight]."

Yet, as no doctor was exclusively dedicated for the hospital MDR-TB centre, physicians were not reliably available for the care that patients demanded during emergent medical conditions. The problem has hampered patients' ability to cope-up with the complications arising from the side effects of the treatment given for MDR-TB. An excerpt taken from a patient with a history of medical emergency illustrates this:

*". . . at one time I was seriously ill and I was brought by car to this centre. When we arrived, there was no doctor. Then I fainted and was near death. He came five hours after he was called and that was when I lost consciousness; at that time, it means that I was dead. . .it would have been good if the doctor was here and I was treated on time and I could tell him about the pains and problems I had. . ."*

[female participant, number one]."

Thus, patients were dissatisfied with the absence of reliable care by a physician during their emergent medical conditions. At hospitals, the views and service preferences of patients is not considered regarding the nutrition support that they received from hospitals. Patients were dissatisfied with the poor quality, inadequate quantity and on the mode of delivery of the nutrition support that they were given; and with the absence of patient involvement in nutrition-related decision-making. The caring practices of caregivers at the treatment follow-up centres (TFCs) was described as disrespectful and uncaring. Patients were dissatisfied with the poor communication and the uncaring behaviour of most caregivers found at TFCs. The next excerpt captured from a patient who discontinued college education to attend to the treatment for MDR-TB clarifies this:

*". . .I was a first-year university student. . .I discontinued my education to be treated and cured from this disease, but I have discovered that only few people provide services with respect. Some of them do not even consider us as human. . .they do not act professionally and sometimes we wait for five to six hours to get the daily medication we need. . .*

[male participant number 13]."

The concept in the above excerpt signals us on the presence of caregivers' attitudes and practices that constitutes an act of stigma against patients with MDR-TB.

Exploration of patients' level of satisfaction with the physical facilities of the MDR-TB treatment centres revealed that patients were satisfied with the cleanness of the open premises of the treatment centre at hospital but dissatisfied with lack of recreation facilities within the premises that was dedicated for patients with MDR-TB. Moreover, patients were dissatisfied with lack of cleanness of the utilities within the centre. Patients experienced that patients' living and shower rooms and the toilets were not cleaned on demand, sometimes making it difficult for patients to utilize them. It was also revealed that patients were not allowed to go out of the premises of the MDR-TB treatment centres, mainly to prevent disease transmission as a result of inadvertent patient movement within the community. In the absence of recreation facilities within the premises of the MDR-TB centre, patients were bored of staying in the centre and the situation was perceived by some patients as confinement as it isolates them from social life. The next excerpt taken from a patient treated as inpatient at a hospital illustrates how a social exclusion that started at home further worsened by the situation at the hospital MDR-TB treatment centres:

*". . . the social life, you cannot live with others. I know what happened to me. . .before I came here, all the neighbours and all family members avoided me. . .. when they brought me to this hospital, here also there was no recreation and that affected me mentally until now, they give me food and you can say that it is a prison for me. It is difficult to be separated from family. It is what God gave me and I did not buy the disease. . .*

[male participant number 14]".

## Socio-economic impact of MDR-TB

Poverty, socio-economic difficulties, and status of available socio-economic support for patients were the subthemes considered under this main theme. Patients experienced that MDR-TB caused a range of social and economic impacts on patients with the disease. These include separation from family and friends because of fear of disease transmission and stigma toward patients with MDR-TB. For some patients, discrimination resulted in termination of employment. This was clarified in the excerpt below:

*". . .I have stopped my job, once they knew that I have MDR-TB, there was a big problem, people did not welcome me and their attitude became negative towards me and also, I did not have the strength to work like before. . .*

[male participant number 2]."

Some of the patients were bread winners for their families and had dependents to take care of. Such patients expressed the desperate condition associated with becoming a patient with MDR-TB. Fifteen of the 18 interviewees (83%) were not engaged in any income generating work at the time of this study. Furthermore, some patients attributed their catching MDR-TB to their poverty and lack of adequate food and prolonged hunger. The next excerpt clarifies this:

*". . .I think it may be from hunger and thirst that I caught the disease because I was employed in daily labour work, and I worked in the deserts where I could not get food and water when I needed. . .*

[male participant number 15]."

## Socio-economic support provided for patients with MDR-TB

Socio-economic support for patients with MDR-TB was part of the clinical care provided by the programme of MDR-TB. At the hospital level, admitted patients were given food and accommodation services. Yet, the nutrition services given for admitted patients was reported to be of poor quality and not quantitatively adequate. Almost all participants were nervous when the issue of nutrition support was presented for discussion. The next excerpt taken from two inpatients treated at the hospital clarified this:

*". . .about the food, it is better not to discuss about it [*male participant number 9*]" . . .;
". . .with respect to food, we are hurt. . . every morning, at lunch time and at dinner, you are served with the same dish, that is all about the food we are given. . .*

[female participant number 1]."

During the outpatient phase of the treatment, patients were given nutrition packages and reimbursed for transport costs that they incurred during their travel to attend to the monthly MDR-TB clinic days at hospitals. It was revealed that the mode of delivery of the monthly nutrition support is not patient centred as it was disbursed at the hospitals, far away from the patients' home areas. Thus, patients incurred an unnoticed cost to transport the nutrition items from the hospitals to their home areas. Likewise, the amount of reimbursement made on transport costs was mentioned to be inadequate, as it lacks a system to cover expenses incurred for local transporters like carts and motorbikes for which formal receipts are not available. Patients perceived that their opinions and views were not solicited by the management of the hospitals to identify available service gaps and address them. Thus, routine planning and implementation of the nutrition support is not aligned towards the needs and service preferences of patients with MDR-TB. An attending caregiver describes the situation in the next excerpt:

*". . .as a physician treating these patients, . . .patients should get a variety of food items and it should be given based on their preferences, . . .they are given food items and the majority of the food items are not body building, . . .they are sources of more of carbohydrates, how does the grain flour help and those foods are given simply because it is food . . .I do not support much of these things because patients should get variety of food and also based on their interest; I have also seen that patients want to select the type of food that they want to eat,. . .,therefore to say that it really builds their body and prevent their body from this disease, even as the science states, it should be protein foods like egg and milk,. . .*

[Male caregiver number 7]."

## Adverse drug reactions (ADRs) and its management

Occurrence and impacts of ADRs, availability of ancillary drugs and status of caregivers' responsiveness in managing ADRs were the constituents explored under this main theme. The group of common antimicrobials used for the treatment of MDR-TB include Levofloxacin/ Moxifloxacin, Bedaquiline, Linezolid, Delamanid, Cycloserine, clofazimine, Ethionamide/ Prothionamide and PAS [30, 31]. These antimicrobials are associated with ADRs among patients treated for MDR-TB [38]. This study revealed that ADRs are major challenges on patients' adherence to the standard treatment given for MDR-TB. All patients who participated in this study had experienced some form of ADRs while on treatment. Participants perceived that the drugs taken for MDR-TB are miserable and it is too difficult for them to take the drugs daily. Vomiting, burning sensation in the body, decreased hearing, blurring of vision, joint pain and exhaustion were the commonly mentioned ADRs. The next excerpt from a patient with history of ADRs elucidates the extent of challenge posed by ADRs:

*". . .once I developed the disease, I tried to tolerate all the burnings. . .I usually ate a piece of sugarcane just to sooth my body. . .I had pain in the joints but the doctor said that it would disappear and I should take it easy, but still it continued and became more severe,. . .*

[male participant number 10]".

Because of the ADRs, the perceived seriousness of MDR-TB by patients was heart-breaking. Patients were pessimistic; while still taking the treatment and some patients were doubtful on their chance of getting cured from the disease. This was captured in the next excerpt:

*"...the disease is deadly...out of five of us that were admitted at the same time to this hospital, only two of us were discharged alive and the other three patients died of the disease...*

[male participant number 4].

Caregivers clarified that substantial number of patients with MDR-TB were repeatedly readmitted to the hospitals due to the repeated occurrence of ADRs. According to caregivers, the hospitals lacked key laboratory tests like the electrolyte and organ function tests which are needed for early diagnosis and prompt management of ADRs. There were incidences of sudden patient deaths which were attributed by caregivers to presumably undiagnosed ADRs. Moreover, ancillary drugs needed to treat ADRs were not consistently available through the programme of MDR-TB. It was also revealed that in some instances, patients' psychological derangements from ADRs caused changes in the patients' behaviour. As families and friends did not have insight into the issue, such changes in the patients' behaviour was usually misunderstood and resulted in patients being discriminated by the family, friends and the wider community.

## Status of integrated care for MDR-TB and HIV/AIDS

It was reported that significant number of patients with MDR-TB were co-infected with HIV/AIDS. An excerpt taken from an attending physician illustrates this:

*"...thirty to forty percent of the patients with MDR-TB who have been receiving treatment with me had both HIV and MDR-TB...and the patients' chance to die is very high...*

[*male caregiver, number 7*]."

Despite the high rate of MDR-TB and HIV/AIDS co-infection, patients affected by both diseases could not get services on both diseases under one roof and by the same caregiver. At the MDR-TB treatment centre, there was no antiretroviral drugs; and the caregivers at the centre did not have the trainings to prescribe antiretroviral drugs. Thus, co-infected patients were referred to other centres for the treatment and follow-up services they needed for HIV/AIDS. It was reported that the co-infected patient group suffered from the inconveniences associated with seeking care for both diseases from different caregivers those located at different centres. Moreover, erratic movement of patients to attend care at different centres was perceived by caregivers as a potential risk factor for disease transmission to the community. Caregivers reported that the majority of deaths were observed among MDR-TB and HIV co-infected patients. When enrolled to the MDR-TB treatment centre, co-infected patients usually revert their attention to the new challenges of coping up with the treatment given for MDR-TB. Thus, patients could not continue adhering to the standard follow-up schedules of HIV/AIDS. As such, there were incidences of sudden deaths among the co-infected patients, which the caregivers attributed to the absence of optimum care and follow-up services for HIV/AIDS at the MDR-TB treatment centre. In the excerpt below, a physician who was treating co-infected patients, desperately narrated the incident of sudden death that occurred to an MDR-TB and HIV/AIDS co-infected patient in the 17th month of patient's treatment for MDR-TB:

*"...a patient was on antiretroviral therapy for seven years and then he caught MDR-TB and was on treatment for MDR-TB for seventeen months. The patient focused on the new problem of MDR-TB and the challenges associated with taking both second-line anti-TB drugs and the antiretroviral drugs. In such scenario, the patient could not continue the routine follow-ups needed to make sure the continued success of the antiretroviral therapy in suppressing the*

*patients' viral load. While hopefully completing his treatment for MDR-TB, one day, the patient suddenly fell on the road and found comatose. Then the patient was brought to this MDR-TB treatment centre, and it was discovered that the patient's CD4 count was only 40 cells per cubic milliliter, which indicated failure of the patient's antiretroviral treatment, and the patient died despite the efforts we made. . . the antiretroviral therapy failure was not diagnosed until the patient was found fell down on the road, and that may have been the possible cause of the patient's death,. . ..*

[*male caregiver, number 8*]."

## Functionality of the MDR-TB programme

Ease of access to service points by patients, system's support to the programme of MDR-TB on service coordination with the community to ensure continuum of care were constituents considered under this main theme. As patients with MDR-TB lacked insight into the programme of MDR-TB, this theme was explored through interviews with caregivers. Caregivers elucidated that patients with MDR-TB were managed following an ambulatory model of care whereby patients are initiated on treatment at hospitals and then linked to the community-based treatment follow-up centres (TFCs) where they continue clinical follow up services. It was revealed that daily patient support through observation of treatment by health workers was not functional beyond the TFCs. Community health extension workers (HEWs) were not engaged in the provision of daily treatment support for patients with MDR-TB. Thus, patients with MDR-TB were forced to attend their daily treatment services at the standard TFCs. The TFCs were described to be far from home areas of patients who came from remote rural areas. Such patients had to incur an unnoticed expense for accommodation at the hometown of TFCs to attend to the daily observed treatment. The situation was described to be difficult for the remote rural patients who cannot afford accommodation fees; and the condition was reported to be negatively impacting patients' adherence to the lengthy treatment given for MDR-TB.

The MDR-TB programme operates on the nationally dedicated programme fund [22]. Yet, the regional health bureau, to which the hospitals are accountable, lacks a strong and uniform system to ensure effective utilisation of programme budgets especially in areas of socio-economic support for patients with MDR-TB. There was no system for addressing the fundamental values and service preferences of patients on patient-centredness, quality and quantity of the nutrition support provided by the programme. The opinions of patients and their clinical caregivers were not used to guide the socio-economic support provided by the programme. Although patients with unequal socio-economic status (SES) needed differing socio-economic support, the programme followed a uniform or a per-capita approach to patients' socio-economic support. The level of support from the management of hospitals to the efforts of clinical care team was described to be weak. For instance, non-clinical staffs of the hospitals were not fully engaged in providing non-clinical services expected of them. Thus, the clinical care team was responsible for facilitating disbursement of nutrition packages and transport costs to patients with MDR-TB. The next excerpt illustrates this:

*"When you think as a physician, as others say, 'your patients'. . .'your patients', you are made responsible for everything that the patients need, the hospital management does not have any concern for this issue. . .there are many things that we facilitate; if water and electricity is discontinued it is the physician who fights, we are logisticians, and finance person as it is we who*

*runs the financial payment. We are pharmacists as we receive and dispense drugs; we are also laboratory technologists as we collect and submit laboratory samples. . .*

[*male caregiver, number 5*]."

Moreover, as the collaboration from the immediate health management units is weak, coordination of patients' linkage to community level TFCs and retrieval of patients who are lost to follow-ups were left to the responsibility of the clinical care team.

## Respiratory MDR-TB infection control practices

Risks of disease transmission at health facilities and to household contacts of patients were constituents of this main theme. Hospital level caregivers reported optimum level of alertness on the risk of respiratory MDR-TB transmission and presence of acceptable infection control practices. Potentially infectious inpatients were kept separately from culture converted ones. There is strict use of personal protections by patients, caregivers, and family caretakers. On the other hand, there was no system for ensuring respiratory MDR-TB infection control at the community and patients' household levels. It was invariably mentioned by patients and their caregivers that the larger community did not have adequate insight into the danger of respiratory transmission of MDR-TB and the challenges associated with its treatment. Thus, MDR-TB was not considered as a serious disease by the community. Instead, patients who experienced the challenges of coping with the disease made efforts to protect the community and their loved ones by refraining from inadvertently mingling with the community. The next excerpt illustrates this:

*". . .Yea, in our village, people do not know much about MDR-TB. They do not know that the disease is difficult to cure. They do not perceive it as a serious disease. . .but because now I know about the disease, I refrain from mingling with people. . .*

[*female participant, number 1*]."

Moreover, efforts were not made by community level health workers to reduce risk of MDR-TB transmission at the household level. It was revealed that caregivers at the TFCs and community health extension workers were not visiting home areas of patients for educating families of patients and making housing arrangements before the patients are sent back to their home. Moreover, family member caretakers were not given respirators as personal protection. Thus, due to lack of the insight and the equipment for personal protection, some of the family caretakers contracted MDR-TB while taking care of family members who had the disease. *The excerpt below taken from a patient on treatment elucidated this*:

*". . .I caught the disease while I was taking care of my husband. I had never had TB before. . . he was my husband. . .while I was taking care of him, I did not know about the disease because both of us did not have this disease before. They put him on the six-month treatment and at that stage, they did not tell us about any precautions, and no advice was given for him also. When he started the treatment, I did not think that the disease transmits and in fact he is my husband, and I could not abandon him because he is sick and I could not go away. If we were told that it transmits, I would have taken care and he also would have taken care of me,. . .*

[*female participant, number 6*]."

Moreover, a nurse caregiver narrated an incident in which four household contacts of a single index case in one family were diagnosed with MDR-TB in which only the father escaped the disease. In a nutshell, the study revealed presence of optimum respiratory MDR-TB infection prevention practices at the hospitals and absence of a functional system for respiratory MDR-TB infection prevention at the community and patients' home levels.

## Discussion

Patients with MDR-TB are satisfied with the compassionate clinical care provided by caregivers at hospitals. Yet, at hospitals, the care and services provided for patients lacked at least two important attributes of quality. First, it is not timely, as there are delays in providing emergent medical care for patients who need to receive it. Second, it is not patient-centred, i.e., the values and preferences of the patient does not guide clinical and non-clinical decision making [21]. Thus, patients were dissatisfied with the absence of reliable care by a physician and with the poor quality, inadequate quantity, and lack of patient-centredness of the nutrition support provided by the hospitals. The poor communication and uncaring practices of caregivers at most of the TFCs signals the presence of stigma from caregivers against patients with MDR-TB. The study revealed that most patients with MDR-TB lived in socio-economic difficulties; and the poor patients depended solely on the socio-economic support provided by the programme. Moreover, the poor patients shared their nutrition with their dependents which resulted in an ongoing inadequate food intake by the poor. Thus, socio-economic hardship hampered patients' ability to cope with MDR-TB and its treatment. Absence of a strong system for monitoring appropriate implementation of patients' socio-economic support furthers available gaps in this area. The result of this study is consistent with extant evidence in that patients with TB are too weak to continue engagement in their routine job activities. Thus, the burden of the direct costs needed for seeking care for the patient and the indirect costs incurred as a result of lost jobs is borne by patient families [22, 32]. Moreover, the finding is consistent with *the theory of the fundamental cause of health inequality* among individuals with low socio-economic status who lack resources like money, knowledge, and beneficial social connections that protect health irrespective of what mechanism is available to avert adverse outcomes of a given disease. By acting through multiple risk factors that involve access to the resources needed to avoid the risks or minimize the consequences of a disease once it occurs, the fundamental social causes of health inequality influences multiple disease outcomes, including mortality, among the socially under- privileged group of the society [33].

Adverse drug reactions (ADRs) from second-line drugs were common among patients included in this study. On the other hand, the hospitals lacked the key laboratory tests needed for early diagnosis and management of the ADRs. It is repeatedly cited that unless they are timely diagnosed and promptly treated, some of the adverse drug reactions like electrolyte abnormalities (hypocalcemia and hypokalemia) can result in lethal outcomes especially among HIV co-infected patients with malnutrition [34]. In this view, the absence of a fully equipped laboratory is a potential risk factor that hampers patients' coping ability with the challenges associated with the treatment given for MDR-TB. Available evidence indicates that MDR-TB and HIV co-infection is associated with poor treatment outcomes [35–37]. As a result, WHO recommends integrated care on MDR-TB and HIV/AIDS especially in countries like Ethiopia that are affected by the dual burden of the two diseases [38–42]. This study revealed, a high level of MDR-TB and HIV co-infection and absence of integrated care that is given under one roof and by the same caregiver. Given that both MDR-TB and HIV need regular clinical and laboratory follow- ups, the absence of integrated care for both diseases have put co-infected patients at difficulty to comply with the treatment and follow-up requirements of both diseases

as it entails them visiting different caregivers located at different centres. Level of support to the programme of MDR-TB by the nearby health management units in engaging community health workers in community education and in assisting patients with MDR-TB through observation of daily patient treatments and in the retrieval of patients lost to follow ups were revealed to be suboptimal. These non-clinical tasks were revealed to be additional burden left to the responsibility of the clinical care team. Absence of optimum respiratory MDR-TB infection control practices at the community and patients' household levels, inadequate community awareness on risk of respiratory MDR-TB transmission and the erratic movement of patients within community to attend to the follow up care and services located far away from their home areas allows continued disease transmission in the community [43].

## Conclusion

This study has identified clinical and socio-economic factors that determine patients' perceived quality of care and their satisfaction with the care given for MDR-TB. The authors believe that, the results of this study can assist evidence-informed decision making in the Oromia Region of Ethiopia and beyond to mitigate factors affecting patients' satisfaction in the programmatic management of MDR-TB.

## Supporting information

**S1 File.**
(DOCX)

## Acknowledgments

The authors express gratitude for the University of South Africa, Department of Health Sciences, and Oromia Health Bureau, health research core-process both for providing us ethical clearances required to conduct the study. We are also thankful to patients with MDR-TB and their caregivers who provided us with the necessary data required to answer the research questions of the study.

## Author Contributions

**Conceptualization:** Mengistu K. Wakjira, Peter T. Sandy, A. H. Mavhandu-Mudzusi.

**Data curation:** Mengistu K. Wakjira.

**Formal analysis:** Mengistu K. Wakjira.

**Funding acquisition:** Mengistu K. Wakjira, A. H. Mavhandu-Mudzusi.

**Investigation:** Mengistu K. Wakjira.

**Methodology:** Mengistu K. Wakjira, Peter T. Sandy, A. H. Mavhandu-Mudzusi.

**Project administration:** Mengistu K. Wakjira, Peter T. Sandy, A. H. Mavhandu-Mudzusi.

**Resources:** Mengistu K. Wakjira.

**Software:** Mengistu K. Wakjira, A. H. Mavhandu-Mudzusi.

**Supervision:** Mengistu K. Wakjira, Peter T. Sandy, A. H. Mavhandu-Mudzusi.

**Validation:** Mengistu K. Wakjira, Peter T. Sandy, A. H. Mavhandu-Mudzusi.

**Visualization:** Mengistu K. Wakjira, Peter T. Sandy.

**Writing – original draft:** Mengistu K. Wakjira.

**Writing – review & editing:** Mengistu K. Wakjira, Peter T. Sandy, A. H. Mavhandu-Mudzusi.

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
