## [Decision Letter · Decision Letter 0]

14 Aug 2022

PONE-D-22-16727Patients’ perceived quality of care and their satisfaction with care given for MDR-TB at referral hospitals in EthiopiaPLOS ONE

Dear Dr. Wakjira,

Thank you for submitting your manuscript to PLOS ONE. After careful consideration, we feel that it has merit but does not fully meet PLOS ONE’s publication criteria as it currently stands. Therefore, we invite you to submit a revised version of the manuscript that addresses the points raised during the review process.

The manuscript is reviewed by independent experts and both have opined that manuscript though it has some merit, requires revision, and I agree with the reviewers. 

We look forward to receiving your revised manuscript.

Kind regards,

Sarman Singh, MD, FRSC, FRCP

Academic Editor

PLOS ONE

Journal Requirements:

"No"

Additional Editor Comments:

The manuscript is reviewed by independent experts and both have opined that manuscript though it has some merit, requires revision. Please see attached reviewer comment files.

Reviewers' comments:

Reviewer's Responses to Questions

**Comments to the Author**

1. Is the manuscript technically sound, and do the data support the conclusions?

Reviewer #1: Yes

Reviewer #2: Partly

2. Has the statistical analysis been performed appropriately and rigorously? 

Reviewer #1: No

Reviewer #2: No

3. Have the authors made all data underlying the findings in their manuscript fully available?

Reviewer #1: No

Reviewer #2: No

4. Is the manuscript presented in an intelligible fashion and written in standard English?

Reviewer #1: Yes

Reviewer #2: Yes

5. Review Comments to the Author

Reviewer #1: • Ethical approval number for collected samples from volunteers should be provided.

• The quality of English writing is good, but some grammatical and structure errors should be revised.

• It is not recommended to write any abbreviation in manuscript title

• Any abbreviation should be written in full name for its first description, thereafter, the abbreviations only are mentioned.

• Common antimicrobials used for MDR-TB treatment of patients should be provided

Reviewer #2: Manuscript seems to be interesting and contributing to the health system in given context. It contributes to the current knowledge of patient's perception and satisfaction about health care services related to MDR TB in ethiopia.

6. PLOS authors have the option to publish the peer review history of their article (what does this mean?). If published, this will include your full peer review and any attached files.

Reviewer #1: **Yes: **Norhan K Abd El-Aziz

Reviewer #2: **Yes: **Prof. Arun Kokane

---

## [Author Response · Author response to Decision Letter 0]

3 Oct 2022

Table of responses to points raised by academic editors/reviewers.

Reviewers’ comments Explanation from the authors

Introduction section 

Write abbreviations in full the first time they are mentioned in the body of the manuscript Whenever they are used, abbreviations are written in full the first time they are mentioned in the body of the manuscript. 

Provide the burden of MDR-TB in Ethiopia This is addressed as per this paragraph: “According to Biadglegne et al 2014, Ethiopia is among the 30 countries in the world described by the World Health Organization as high burden for TB and MDR-TB. Thus, the government rolls out community-based programmatic management of drug-resistant TB across all its provinces.” 

Ethical approval number for collected samples from volunteers should be provided. Participant information sheet and consent form used during participant selection is attached 

Revised grammatical and structural errors Revisions are made to correct any grammatical and structural errors

Duplication of definitions of perceived quality and patient satisfaction in terms of “patient’s expectations and their perceptions on the quality of services”. This is revised and clarified throughout the body of the manuscript 

Materials and methods section 

Letter of permission from the ethics committee Ethical approval from institutional board and letter of permission from relevant health department’s research directorate is obtained and it is uploaded

The type of qualitative approach used to conduct the study An inductive phenomenological approach was used in this study to understand the contextual lived experience of patients with MDR-TB and their caregivers 

Subheadings should identify “study setting” and “study period” This comment is addressed in the materials and methods section of the manuscript

How were participants approached for interview (face-to-face; telephone, mail.) Interviews were made by the principal investigator (MK) through face-to-face interviews with patients with MDR-TB. This is mentioned under data collection section of the manuscript.

Presence of a non-participant during individual interviews A trained note-taker was taking notes on the feelings and experiences of the study participants. This is mentioned in the data collection section of the manuscript. 

Selection criteria Information adults aged 18 years and above who can consent for the interview were eligibility criteria. Information was not obtained from those aged under 18 years!

Who were the interviewer The principal investigator (MK Wakjira) conducted the in-depth interview with patients and their caregivers. The principal 

Was the interview audio taped and notes taken Interview data were audio recorded and transcribed verbatim. Notes were taken during interviews where data were captured on the feelings and experiences of the study participants. Data from multiple sources were used to make sure that emerging themes are established based on converging different sources of data or different perspectives of the study participants.

Description of the study sample A total of eighteen adult patients with MDR-TB; 9 (50%) female and 9 (50%) male patients were included in the study. Three of the 18 (17%) participants openly disclosed that they were MDR-TB and HIV co-infected. Moreover, eleven caregivers for MDR-TB (3 physicians and 8 nurses) participated in the in-depth interviews. 

Language used to for the interview The patients mother tongue language (Afan Oromo and Amharic) was used for the interview. As an Ethiopian national, the interviewer (MK Wakjira) is proficient in both languages and language. 

Duration of each interview Each individual interview was conducted for about 30-40 minutes (this is indicated in patients information sheet and consent form uploaded with the manuscript.

Was data saturation discussed Yes! As discussed under “study population and participant selection” section, the number of participants of the interviews was determined by category saturation, the point at which the interviews did not reveal new data relevant to the objectives of the study

Are transcripts returned to participants Yes. Member checking was employed with each participant in which summarized interview data were presented to each interviewee and corrections sought to understand if what the researcher captured matches the intentions and opinions of the participant. Transcripts were also given to some literate participants, and they were asked if what the researcher captured matches their opinions. Moreover, two debriefers who had experience in social research were located and their frequent feedbacks were used to enhance accuracy of the construct under scrutiny.

Data Availability Authors declare that data underlying the findings of this study is available without restriction. The whole verbatim transcript of the qualitative data is uploaded

Results section 

Include common antimicrobials used to treat MDR-TB Included under the section “Adverse drug reactions and its management”

Abbreviations could be given at the end of the manuscript or somewhere while being used for the first time within the text for letters like P-participants This is addressed starting from the “abstract page” of the manuscript.

For letters like P-participants and C.G-caregivers a footnote I used to illustrate them on the same page where the letters are used as abbreviations.

Provide common antimicrobials used in the treatment of MDR-TB List of common antimicrobials used in the treatment of MDR-TB is provided under the section dealing with “adverse drug reactions from drugs used in he treatment of MDR-TB”

Begin the results section with major and sub-themes This comment is addresses in the result section

There were repetitions pertaining to the nutritional and socio-economic status in the health system theme as wells in the socio-economic theme. Concerns from participants can be captured as an excerpt. The repetitions are revised and corrected. The concerns and views of patients with MDR-TB is clearly raised and it is also indicated that there is no feedback from the system on the concerns and views of patients with MDR-TB. 

Over-emphasis of acquiring MDR-TB among household contacts. Add a brief write up on this A brief writes up including excerpt is provided in this section 

Verbatim for respiratory MDR-TB infection control is missing An excerpt illustrating the risk of and situation of respiratory MDR-TB infecting is included

The study is on patients’ perception of quality of care and satisfaction, inclusion of suitable verbatim might be added An excerpt is added to illustrate patients’ perception of the quality of care they received at hospitals and at the community level y treatment follow up centers. Excerpt is also added to make similar illustrations on the non-clinical (socio-economic) services and support that patient got from the programme of MDR-TB

Reports on satisfaction to avoid reporting bias This is clearly reported under the title of “Patients’ perceived quality of care and their satisfaction with the care given for MDR-TB”- At hospital level, the communication between patients and their caregivers was revealed to be optimum. Thus, the majority of patients were satisfied with the smooth communication and empathetic clinical care they received at hospitals….

 

Discussion 

Could be better organized along with the flow of the themes This comment is addressed in the discussion session. The discussions are reorganized along with the flow of the themes in the study.

---

## [Decision Letter · Decision Letter 1]

17 Jan 2023

Patients’ perceived quality of care and their satisfaction with care given for MDR-TB at referral hospitals in Ethiopia

PONE-D-22-16727R1

Dear Dr. Wakjira,

We’re pleased to inform you that your manuscript has been judged scientifically suitable for publication and will be formally accepted for publication once it meets all outstanding technical requirements.

Kind regards,

Sarman Singh, MD, FRSC, FRCP

Academic Editor

PLOS ONE

Additional Editor Comments (optional):

Reviewers' comments:

Reviewer's Responses to Questions

**Comments to the Author**

1. If the authors have adequately addressed your comments raised in a previous round of review and you feel that this manuscript is now acceptable for publication, you may indicate that here to bypass the “Comments to the Author” section, enter your conflict of interest statement in the “Confidential to Editor” section, and submit your "Accept" recommendation.

Reviewer #2: All comments have been addressed

2. Is the manuscript technically sound, and do the data support the conclusions?

Reviewer #2: Yes

3. Has the statistical analysis been performed appropriately and rigorously? 

Reviewer #2: Yes

4. Have the authors made all data underlying the findings in their manuscript fully available?

Reviewer #2: Yes

5. Is the manuscript presented in an intelligible fashion and written in standard English?

Reviewer #2: Yes

6. Review Comments to the Author

Reviewer #2: Most of comments are addressed by the author. The study is drafted satisfactorily and may be accepted for publication.

7. PLOS authors have the option to publish the peer review history of their article (what does this mean?). If published, this will include your full peer review and any attached files.

Reviewer #2: **Yes: **Prof. Arun Kokane
